# Technology-Assisted Collaborative Care Program for People with Diabetes and/or High Blood Pressure Attending Primary Health Care: A Feasibility Study

**DOI:** 10.3390/ijerph182212000

**Published:** 2021-11-15

**Authors:** Pablo Martínez, Viviana Guajardo, Víctor E. Gómez, Sebastián Brandt, Wilsa Szabo, Gonzalo Soto-Brandt, Maryam Farhang, Paulina Baeza, Solange Campos, Pablo Herrera, Graciela Rojas

**Affiliations:** 1Departamento de Psiquiatría y Salud Mental, Hospital Clínico Universidad de Chile, Santiago 8431617, Chile; pablo.alberto.martinez.diaz@usherbrooke.ca (P.M.); viviguajardo@gmail.com (V.G.); victorgomezp@gmail.com (V.E.G.); farhangmaryam.psy@gmail.com (M.F.); 2Millennium Institute for Depression and Personality Research (MIDAP), Santiago 7820436, Chile; wilsaszabo@hotmail.com; 3Millennium Nucleus to Improve the Mental Health of Adolescents and Youths (Imhay), Santiago 8380455, Chile; 4Escuela de Psicología, Facultad de Humanidades, Universidad de Santiago de Chile, Santiago 9170197, Chile; 5Psicomedica, Clinical & Research Group, Santiago 7500710, Chile; 6Servicio de Psiquiatría, Hospital El Pino, Santiago 8050000, Chile; 7Programa de Doctorado en Psicoterapia, Facultad de Ciencias Sociales, Pontificia Universidad Católica de Chile, Santiago 7800284, Chile; 8Society for Psychotherapy Research, Krestwood, KY 40014, USA; 9Facultad de Medicina, Universidad de Chile, Santiago 8380453, Chile; 10Escuela de Psicología, Facultad de Ciencias Sociales, Universidad de Chile, Santiago 7800284, Chile; sebastian.brandt@ug.uchile.cl (S.B.); pabloherrerasalinas@gmail.com (P.H.); 11Programa de Salud Mental, Escuela de Salud Pública, Universidad de Chile, Santiago 8380453, Chile; bsoto@uchile.cl; 12Programa de Magíster de Salud Mental y Psiquiatría Comunitaria, Escuela de Salud Pública, Universidad de Chile, Santiago 8380453, Chile; p.baezatorres@gmail.com; 13Escuela de Enfermería, Facultad de Medicina, Pontificia Universidad Católica de Chile, Santiago 7820436, Chile; scamposr@uc.cl; 14Millennium Nucleus of Social Development, Santiago 8330015, Chile

**Keywords:** depression, chronic disease, disease management, primary health care, information technology, feasibility studies

## Abstract

The comorbidity of depression with physical chronic diseases is usually not considered in clinical guidelines. This study evaluated the feasibility of a technology-assisted collaborative care (TCC) program for depression in people with diabetes and/or high blood pressure (DM/HBP) attending a primary health care (PHC) facility in Santiago, Chile. Twenty people diagnosed with DM/HBP having a Patient Health Questionnaire-9 score ≥ 15 points were recruited. The TCC program consisted of a face-to-face, computer-assisted psychosocial intervention (CPI, five biweekly sessions), telephone monitoring (TM), and a mobile phone application for behavioral activation (CONEMO). Assessments of depressive symptoms and other health-related outcomes were made. Thirteen patients completed the CAPI, 12 received TM, and none tried CONEMO. The TCC program was potentially efficacious in treating depression, with two-thirds of participants achieving response to depression treatment 12 weeks after baseline. Decreases were observed in depressive symptoms and healthcare visits and increases in mental health-related quality of life and adherence to treatment. Patients perceived the CPI as acceptable. The TCC program was partially feasible and potentially efficacious for managing depression in people with DM/HBP. These data are valuable inputs for a future randomized clinical trial.

## 1. Introduction

Depressive disorders and chronic diseases are among the leading causes of disability worldwide [1]. Comorbidity is the norm rather than the exception [2], as up to two-thirds of depressed patients in primary healthcare (PHC) may present a chronic disease [3]. Such a frequent phenomenon has been linked to poor clinical outcomes and functional status [4,5], less adherence to treatment [6], higher health-care expenses, and mortality [7,8,9].

The effectiveness of interventions for depressive disorders in people with chronic illnesses has not been sufficiently explored. However, the collaborative care (CC) model seems to be the most effective in reducing depressive symptoms and promoting adherence to the treatment of chronic diseases [10,11,12]. This complex intervention model incorporates shared responsibilities between mental health specialists and general health teams for integrated health care [13]. Though promising, evidence from developed countries suggests that successful experiences required firm organizational commitments [14,15]. Furthermore, its implementation and sustainability may be challenging in developing countries [16].

Brief, evidence-based psychological interventions are essential in the CC model [13,17] and in managing chronic diseases [18,19]. However, treatment adequacy may be strikingly low in PHC [20,21]. The use of information technologies may improve the dissemination and quality of evidence-based interventions. For instance, therapist-supported computer-assisted psychological interventions may effectively relieve depressive symptoms in PHC populations [22]. Up to date, no brief computer-assisted psychological interventions have been tested to manage depression in people with chronic illnesses attending PHC.

Chile, a Latin American country, has made significant efforts to integrate mental health care into PHC [23,24,25,26] and develop health promotion and prevention strategies to control the burden of disease attributable to chronic diseases [27]. Periodically updated clinical guidelines direct the management of depression, diabetes, and high blood pressure, and the access, opportunity, and financial coverage for the care of these health conditions are guaranteed by law. However, the frequent and complex scenario representing the comorbidity between these diseases is not considered [28,29]. Thus, there is a need for more evidence of practices contributing to this direction.

The objective of this study was to evaluate the feasibility of “I Take Care and I Feel Better” for depression in people with diabetes and/or high blood pressure who attend PHC.

## 2. Materials and Methods

### 2.1. Study Design

This is a feasibility study to inform the conduction of a future clinical trial, which focused on the following key areas: acceptability, demand, and potential efficacy. A mixed-methods design combining a quantitative and a qualitative component was used. The first component was a quasi-experimental, uncontrolled before-and-after study conducted to assess the feasibility of a technology-assisted CC program. Afterward, a qualitative nested study allowed a thorough exploration of patients’ experiences with a computer-assisted psychosocial intervention.

### 2.2. Participants

People between 18 to 79 years of age, diagnosed with diabetes and/or high blood pressure in a PHC clinic in Santiago, Chile, serving a low and middle-income population. These individuals were eligible if they scored 15 or higher on the Patient Health Questionnaire 9-item (PHQ-9). The study excluded people who: had alcohol use problems (i.e., Alcohol, Smoking and Substance Involvement Screening Test (ASSIST) score of 27+); were in treatment for bipolar disease or schizophrenia; received psychological attention in the last month, or had a scheduled session for the next 30 days; could not read or write; had a severe cognitive, visual, and/or hearing impairment; did not have the digital understanding/literacy to use an application on a smart mobile phone; or, people who were pregnant or lactating.

### 2.3. Procedures

Trained recruiters approached potential participants for eligibility in the study site waiting rooms. After eligibility assessment with a brief, digitized screening questionnaire, written informed consent was obtained from the patient. Subsequently, participants were reassessed a week later to confirm the persistence of depressive symptoms (i.e., PHQ-9 score ≥ 15 points) and eligibility. During this assessment, participants with confirmed eligibility underwent the baseline evaluation and were assigned to the technology-assisted CC program. Finally, participants were contacted by the study psychologists to initiate treatment a week later.

### 2.4. Intervention

“I Take Care and I Feel Better” is a technology-assisted CC program that included the following components.

#### 2.4.1. Computer-Assisted Psychosocial Intervention

Carried out at the study site, delivered in-person, individually, by study psychologists during five biweekly sessions (30–45 min each), plus one or more unstructured reinforcement sessions, as per patient request. The first five sessions were manualized and highly structured (Table 1), with computer-assisted activities in an attractive audiovisual format. The intervention combined psychoeducation with cognitive-behavioral, problem-solving, and behavioral activation techniques and motivational interviewing principles. A personal care plan was reviewed at the end of each session, and reinforcement cards synthesizing the sessions’ contents were delivered. The study psychologists received 12 h of training, and telephone supervision sessions, upon request with a psychiatrist. Supervision sessions emphasized discussion of the most complex cases or resistant cases.

#### 2.4.2. Telephone Monitoring

Provided by the study social worker weekly for 10–15 min over seven weeks, plus an eighth, final, monitoring session fifteen days after the seventh monitoring. This component aimed to assess the clinical progress and promote treatment adherence (i.e., reception of medical care, consumption of antidepressants, and the use of the mobile phone application) in patients.

#### 2.4.3. Mobile Phone Application

CONtrol EMOcional (CONEMO) is a self-guided (i.e., minimally supported) mobile phone application. The CONEMO intervention is delivered three times a week for six weeks, for a total of 18 sessions. CONEMO had the objective of behaviorally activating each patient, motivating them to carry out pleasant and/or healthy activities, and fulfill responsibilities and/or daily tasks. This intervention was piloted as an independent treatment for depressive symptoms of outpatients with diabetes and/or hypertension in Brazil and Peru, demonstrating acceptability and potential efficacy [30]. In the present study, psychologists were in charge of introducing and installing CONEMO to each patient during the first session of the computer-assisted psychosocial intervention.

### 2.5. Measures and Covariates

Data collection was in charge of the recruitment team. The baseline evaluation was conducted in person or by telephone. Participants completed a telephone follow-up assessment 12 weeks after assignment to the technology-assisted CC program. In both evaluations, outcomes assessors entered the information directly into a digitized questionnaire. Additionally, difficulties understanding questionnaires were documented. Demographics and personal and family health history were recorded at baseline. Data recorded at baseline and follow-up assessment included: depressive symptoms with the PHQ-9; skills for solving social problems with the Social Problem-Solving Inventory—Revised Short Form (SPSI-R: S); health-related quality of life with the Short Form Health Survey (SF-12); use of health-care services; and consumption/adherence to medications for depression, diabetes and/or high blood pressure. Furthermore, at follow-up assessment, an ad-hoc questionnaire was administered to assess the acceptability of the computer-assisted psychosocial intervention.

The PHQ-9 is a self-report questionnaire for evaluating depressive symptoms consisting of 9 questions on a Likert-type scale, with a total score range from 0 to 27. PHQ-9 scores of 5, 10, 15, and 20 indicate mild, moderate, moderately severe, and severe depressive symptoms, respectively [31]. In the original validation study, a cut-off score of 10 or more had sensitivity and specificity of 88% for major depression [31]. The PHQ-9 has been validated in people who consult in Chilean PHC [32].

The SPSI-R: S is a self-report questionnaire for evaluating social problem-solving skills consisting of 25 questions on a Likert-type scale, with a score range of 0 to 20 for each of its five subscales [33]. Higher SPSI-R: S scores reflect better social problem-solving skills. Only the Positive Problem Orientation and Rational Problem-Solving Subscales of the SPSI-R: S were used in this study.

The SF-12 is a self-report questionnaire for evaluating health-related quality of life that consists of 12 questions on a Likert-type scale [34]. This instrument has Physical and Mental Component Summary Scores that synthesize information from eight dimensions. Higher scores on a 0 to 100 scale reflect better health-related quality of life [34].

The questionnaire on the use of health-care services was adapted from the Chilean National Socioeconomic Characterization Survey (CASEN), registering general medical consultations or regular health-care visits, emergency visits in PHC and/or hospitals, specialist medical consultations, specialist mental health-care visits and hospitalizations, during the last three months, indicating the total number of consultations or health-care visits received (or days of hospitalization).

An ad-hoc self-report questionnaire having four dichotomous items was used to evaluate consumption/adherence to medications for depression, diabetes and/or high blood pressure, with negative attitudes indicating non-adherence to treatment.

Lastly, the ad-hoc self-report questionnaire to assess acceptability consists of 12 questions on a Likert-type scale. The construction of this instrument was based on the theoretical acceptability framework of Sekhon et al. [35]. This framework identifies seven dimensions of acceptability: affective attitude, burden, perceived effectiveness, ethicality, intervention coherence, opportunity costs, and self-efficacy. For their interpretation, the scores obtained were transformed into percentages (0% to 100%), a higher percentage reflecting better acceptability of the intervention.

The primary study outcome was response to depression treatment, defined as a reduction of 50% or more in the PHQ-9 score in the 12 weeks follow-up assessment compared to baseline values [31]. The secondary outcomes were changes observed at the follow-up assessment compared to the baseline evaluation in the PHQ-9, the SPSI-R: S, the SF-12 scores, the use of health-care services, and consumption/adherence to medications for studied health conditions. Scores on the ad-hoc acceptability questionnaire at follow-up assessment were also considered.

For the qualitative component, semi-structured post-intervention interviews were conducted with patients who attended at least three sessions of the computer-assisted psychosocial intervention. The semi-structured interviews were conducted by members of the research team in health-care rooms at the study site and lasted approximately one hour. The audio of each interview was recorded on a mobile device. The following topics were addressed: satisfaction with the intervention; positively valued aspects; aspects to improve; differences from usual treatment; impressions on the duration of the intervention; opinions on the graphic aspects involved; and learning achieved.

### 2.6. Data Analysis

Descriptive statistics were calculated for recruitment data, sample characterization, use and adherence to the technology-assisted CC program, and the computer-assisted psychosocial intervention acceptability. Before-and-after data were analyzed to evaluate the potential efficacy of the technology-assisted CC program. To this end, t-tests for normally distributed data and Wilcoxon signed-rank test for non-normally distributed data were used. Regarding dichotomous variables, McNemar’s test was used. Statistical analyzes were assisted with Stata/MP 14.0 software [36].

For the nested qualitative component, interviews were transcribed verbatim and coded by a research team member. Through the constant comparison method, categories were induced according to the precepts of the Grounded Theory [37]. The analysis of qualitative data was assisted by the ATLAS.ti 8.0 software (ATLAS.ti Scientific Software Development GmbH, Berlin, Germany).

### 2.7. Ethics Approval

Ethics approval for this trial was granted by the Scientific Ethics Committee of the Clinical Hospital of the Universidad de Chile on 6 June 2018 and was recorded on approval act No. 36. Additionally, the Quality Committee of the participating primary health care center reviewed the project, granted its approval on 29 April 2019.

## 3. Results

### 3.1. Recruitment

In one month, ten recruitment days were carried out, lasting approximately three hours each. The recruitment team approached 317 subjects for initial eligibility assessment, of whom 28 met the eligibility criteria (8.8%). One week later, 25 subjects were reassessed with the PHQ-9; 20 of these had 15 or more points in this questionnaire, being assigned to the technology-assisted CC program (Figure 1).

### 3.2. Characterization of the Sample

The ages of the 20 participants ranged from 41 to 75 years of age, with an average of 61.3 years of age (standard deviation (SD) = 9.3 years). Two of the participants identified themselves as men and 18 as women. Twelve completed secondary education or less, four completed or incomplete higher technical education, and four completed university education. Only four subjects reported belonging to an ethnic minority (Mapuche). Considering that the participants could report more than one occupation, most reported being retired (*n* = 11) or in household chores (*n* = 9), while six indicated they were working for a living. The most frequent marital status was married (*n* = 9), followed by widower (*n* = 4). Most participants lived with their partner (*n* = 13) and/or with their children (*n* = 12). Table 2 shows complementary personal and family health information.

### 3.3. Use and Adherence to the Components of the Technology-Assisted Collaborative Care Program

Fourteen patients received at least one session of the computer-assisted psychosocial intervention for an average of 14 weeks. Thirteen of these completed the five biweekly intervention sessions, and eleven had at least one unstructured reinforcement session. Among those who received at least one session of the computer-assisted psychosocial intervention, twelve could be monitored by telephone, with an average of seven telephone monitoring sessions per patient (SD = 1.4). In none of the patients could CONEMO be installed as an adjunct intervention. Reasons indicated by the study psychologists were the low digital literacy declared in session by the patients, the lack of access to a free mobile data plan, the low added value, and the more significant perceived burden of receiving CONEMO as an adjunct to computer-assisted psychosocial intervention and telephone monitoring.

### 3.4. Acceptability of the Computer-Assisted Psychosocial Intervention

The computer-assisted psychosocial intervention achieved a global acceptability of 92.3%. In four sub-scales, the subjects indicated the maximum possible acceptability for the intervention (affective attitude, perceived effectiveness, self-efficacy, and intervention coherence). There were scores other than the maximum possible (100%) on the ethics sub-scales (95.8%), burden (80.6%), and opportunity costs (69.4%).

### 3.5. Potential Efficacy of a Technology-Assisted Collaborative Care Program

#### 3.5.1. Depression

Response to depression treatment was achieved by twelve of the study participants. There were statistically significant differences in the means before (M = 20.5, SD = 4.0) and after the intervention (M = 9.3, SD = 5.4) in the PHQ-9, with a decrease of 11.2 average points (t (17) = 7.5, *p* < 0.001) (Figure 2). Participants manifested difficulties in completing the PHQ-9, noting problems attributing somatic symptoms to their mood or physical health conditions. For this reason, a secondary analysis considered only cardinal symptoms of depression, evaluated with the first two items of the PHQ-9 (i.e., PHQ-2, range 0 to 6 points) [38]. This analysis identified a statistically significant reduction in the means before (M = 5.3, SD = 1.0) and after the intervention (M = 2.3, SD = 1.3) in the PHQ-2, on the order of 3 average points (t (16) = 6.9, *p* < 0.001).

#### 3.5.2. Social Problem-Solving Skills

There were no statistically significant differences in the means before (M = 11.4, SD = 4.9) and after the intervention (M = 13.1, SD = 3.6) in the Positive Problem Orientation subscale (t (15) = 1.3, *p* = 0.219). No statistically significant differences were identified in the before (M = 8.7, SD = 5.3) and after intervention (M = 11.6, SD = 3.3) means in the Rational Problem Solving sub-scale (t (16) = 1.9, *p* = 0.071).

#### 3.5.3. Health-Related Quality of Life

There were no statistically significant differences in the before (M = 33.2, SD = 17.7) and after intervention (M = 34.4, SD = 17.8) means in the SF-12 Physical Component Summary Score (t (17) = 0.4, *p* = 0.729). Statistically significant differences were observed in the before (M = 30.7, SD = 18.0) and after intervention (M = 48.1, SD = 15.3) means in the SF-12 Mental Component Summary Score, for an increase of 17.4 points average (t (17) = 3.9, *p* = 0.001).

#### 3.5.4. Use of Health-Care Services

The proportion of cases that reported having received consultations or health-care visits in the last three months before and after the intervention exhibited a trend towards statistically significant differences (McNemar’s test χ^2^ = 4.50, exact *p* = 0.070), decreasing from twelve to six of seventeen subjects. When comparing the medians of the number of consultations or health-care visits in the last three months before (Mdn = 3.5) and after the intervention (Mdn = 0), statistically significant differences were found (Z = −3.27, *p* = 0.001).

#### 3.5.5. Consumption and Adherence to Medications

There were no significant differences in the proportion of cases before and after the intervention that reported consuming medications for depression, diabetes and/or high blood pressure. A statistically significant post-intervention increase in the proportion of cases adhering to medications for diabetes and/or arterial hypertension (% pre-intervention = 23.5%, vs.% post-intervention = 76.5%; McNemar’s test χ2 = 7.36, exact *p* = 0.012) was detected. No changes were observed in the proportion of cases adhering to antidepressant medications.

### 3.6. Experiences of Patients with the Computer-Assisted Psychosocial Intervention

Thirteen patients that participated in the technology-assisted CC program and received at least three sessions of the computer-assisted psychosocial intervention were part of the nested qualitative component. Data analysis led to the emergence of the seven categories detailed below.

#### 3.6.1. Satisfaction with the Intervention

Perception of liking or disliking the computer-assisted psychosocial intervention. The patients stated that they were satisfied with this component of the technology-assisted CC program. Illustrated with the following patients’ quotes: “It was a pleasant experience”, “It was good [the intervention], I did not find it boring”.

#### 3.6.2. Aspects Positively Valued

Specific aspects of the computer-assisted psychosocial intervention were valued as beneficial by the patients: the closeness with the psychologist, the illustrative quality of the intervention’s visual material, and its ease of understanding. Patients’ quotes: “I felt welcomed [by the psychologist]”, “I liked the pictures [slides]”, “It was easy for me to understand the contents [of the intervention]”.

#### 3.6.3. Aspects to Improve

References to activities or contents of the computer-assisted psychosocial intervention that need adjustment: to allow for more time to complete the health-care plan during sessions, to increase the duration of the intervention (i.e., a greater number of sessions), and to provide more variability and familiarity with the examples proposed in the sessions. Patients’ quotes: “It was difficult for me to understand the care plan”, “The treatment was a little short”, “Some examples are not well understood”.

#### 3.6.4. Differences from the Usual Treatment

Comparisons with other psychological, psycho-educational, or medical interventions previously received by patients. These mentions revealed a marked preference for the current intervention regarding empathy of the psychologist, flexibility in the personal health-care plan, and care continuity. Patients’ quotes: “I felt more listened to in this therapy”, “I did not feel compelled to follow instructions so rigid that they give me at the controls”, “The sessions were more frequent, one did not lose track”.

#### 3.6.5. Extension

Specific aspects of the time and/or frequency of the sessions of the computer-assisted psychosocial intervention, highlighting the perception of more frequent sessions than usual in the context of a brief intervention. Mainly, there are mentions of the duration, frequency, and time of the intervention. Patients’ quotes: “My therapy was short”, “Before they used to call me once a month with luck”, “The sessions were longer than usual.”

#### 3.6.6. Graphic Aspects

Linked to the visual, pedagogical, and graphic elements of computer-assisted psychosocial intervention, as well as the support materials provided. They are described as a positive element, which facilitated the understanding of the contents. Patients’ quotes: “The pictures [slides] were very clear”, “I liked that we had a summary, one forgets things”, “The characters of the slides were very nice, they allowed us to understand better”.

#### 3.6.7. Learning Achieved

Explicit references to new knowledge or the different understanding of some aspects reviewed in the intervention sessions were considered, highlighting education in chronic diseases, the relationship between self-care and mood, and the resolution of social problems. Patients’ quotes: “I did not understand how diabetes worked until I saw the explanation in the slides”, “I learned to better solve my problems”, “It is clear to me that how I care affects how I feel”.

## 4. Discussion

### 4.1. Principal Results

The technology-assisted CC program for depression in people with diabetes and/or high blood pressure who attend PHC (“I Take Care and I Feel Better”) was partially feasible to implement. High adherence and acceptability to the computer-assisted psychosocial intervention and telephone monitoring were evidenced. However, the mobile phone application (CONEMO) could not be tested. “I Take Care and I Feel Better” demonstrated potential efficacy in the treatment of depression, with two-thirds of study participants achieving response to depression treatment. Additionally, statistically significant decreases in depressive symptoms and the median of health-care visits, along with statistically significant increases in the mental health-related quality of life and the proportion of cases adhering to medications for diabetes and/or high blood pressure, were observed. The patients perceived the computer-assisted psychosocial intervention as acceptable and satisfactory, highlighting its easiness of understanding, the pedagogical quality of its computer-assisted and supporting materials, along a learning experience focused on the knowledge and management of diseases. Complementarily, patients, suggested increasing the frequency of sessions.

### 4.2. Comparison with Prior Work

Cognitive-behavioral approaches have been previously tested with great success in managing depression in populations with chronic diseases [39,40]. Such brief, evidence-based psychological interventions are an essential component of effective CC programs [13,17]. In the present study, trained psychologists provided a face-to-face, computer-assisted psychosocial intervention at the PHC clinic. The extant literature reports that these interventions have been chiefly provided through a non-face-to-face modality or with little therapist support [22]. Furthermore, only one study included a face-to-face computer-assisted psychological intervention in a CC program in PHC, demonstrating its benefits in patients with anxious and/or depressive symptoms [41]. Thus, our study would be the first to extend this research to depressed people with diabetes and/or high blood pressure.

In addition, the “I Take Care and I Feel Better” program considered two essential tasks for the implementation of CC programs: care manager follow-up and psychiatric consultation [17,42]. These components have been associated with a greater likelihood of achieving improvement in depression [42]. The first of these tasks was conducted by telephone, intensively for an average of seven weeks, providing a convenient, relatively inexpensive, and valuable opportunity to maintain updated patients’ registries. This component allowed to triangulate information between different health-care personnel involved in patient management. Furthermore, the second component may enhance the integration of specialized behavioral health into PHC to manage depressive disorders, providing regular opportunities to discuss patients who are not responding to treatment [17]. Compared to previous studies aimed at integrating physical and mental health care [43,44,45], the psychiatric consultation herein implemented was less intensive and structured and provided over the phone upon the psychologist’s request. The studies mentioned above, conducted in depressed cancer patients, integrated the PHC team with specialized physical and mental health care, achieving better outcomes for depression regardless of cancer prognosis [43,44,45].

CONEMO—a mobile phone application that was the CC program’s final component —was not feasible for implementation. Study psychologists discussed some of the reasons were low levels of digital literacy, the scarce use of digital devices in the patient population, and the perceived burden of receiving an additional intervention. These results are different from those obtained in pilot studies in similar contexts in other Latin American countries [30]. These experiences suggested that patients would be able to use the mobile phone application and that this would support reducing patients’ depressive symptoms [30]. However, nurses dedicated 45 to 60 min to introduce CONEMO during patients’ first appointment in these pilot studies. Moreover, nurses regularly contacted patients in the following weeks to positively reinforce CONEMO use [30]. Such instructions and supports were positively valued by the patients [46]. In our study, the lack of a dedicated support resource and the strong emphasis placed by psychologists upon the computer-assisted psychosocial intervention may have undermined the understanding and willingness of patients to use CONEMO.

### 4.3. Limitations

Our study has a series of important limitations typical of a pilot experience, which must be considered when conducting a definitive clinical trial. First, an uncontrolled before-and-after design was used. Such study design restricts the inference of causality as detected differences may be attributable to intervening variables, regression to the mean, or the effect of time [47]. These limitations are understandable for a feasibility study with the objective to assess whether the intervention could be carried out [48]. It should be noted that patients reported problems in attributing somatic symptoms to their mood or physical health conditions, suggesting that the use of self-report instruments could limit the objective data collection in this population. The use of instruments administered by trained clinicians may be an alternative, although this option would substantially increase the costs and feasibility of the study procedures. Additionally, the qualitative study was limited to exploring patients’ experiences regarding the computer-assisted psychosocial interventions without considering the patients’ views on telephone monitoring nor the mobile phone application. Of particular interest would have been obtaining specific patient information on the relevance and/or added value of telephone monitoring and the failure to implement CONEMO to complement the evaluation of the technology-assisted CC program.

### 4.4. Implications

Considering the objectives of this study, its results, and methodological limitations, it may be judged that the technology-assisted CC program warrants a definitive clinical trial to determine effectiveness. We anticipate that significant efforts will be necessary to meet a sufficient sample size and statistical power, given that about 5% of the subjects who received an initial evaluation were finally recruited for the study. The high levels of adherence found in the technology-assisted CC program hold promise for treatment integrity in the definitive clinical trial. On the other hand, considering patients’ experiences and difficulties, a future study should increase the frequency and number of sessions, incorporating the contents of behavioral activation to replace CONEMO and reinforcing the use of problem-solving techniques. The mobile phone application should be piloted as a stand-alone intervention with appropriate support personnel—as in the pilot experiences in Brazil and Peru [30]. Finally, the promising results of “I Take Care and I Feel Better” suggest that a CC program of these characteristics could effectively manage depression in patients with moderate to severe symptoms, and eventually promote adherence to diabetes and/or high blood pressure medications. The definitive clinical trial will assess the long-term outcomes of this technology-assisted CC program (up to 12 months after recruitment). We may expect that if adherence to the treatment of chronic diseases is sustained over time, changes in indicators of physical health (e.g., levels of blood pressure) might be observed.

## 5. Conclusions

“I Take Care and I Feel Better”, a technology-assisted collaborative care program for depression in people with diabetes and/or high blood pressure, was partially feasible to implement in primary health care, being acceptable to patients, and potentially efficacious in achieving response to depression. A refined version of the intervention should be evaluated in a randomized clinical trial with sufficient statistical power to provide evidence of the effectiveness of this collaborative care program in the management of depression in people with diabetes and/or high blood pressure in primary health care.

## Figures and Tables

**Figure 1 ijerph-18-12000-f001:**
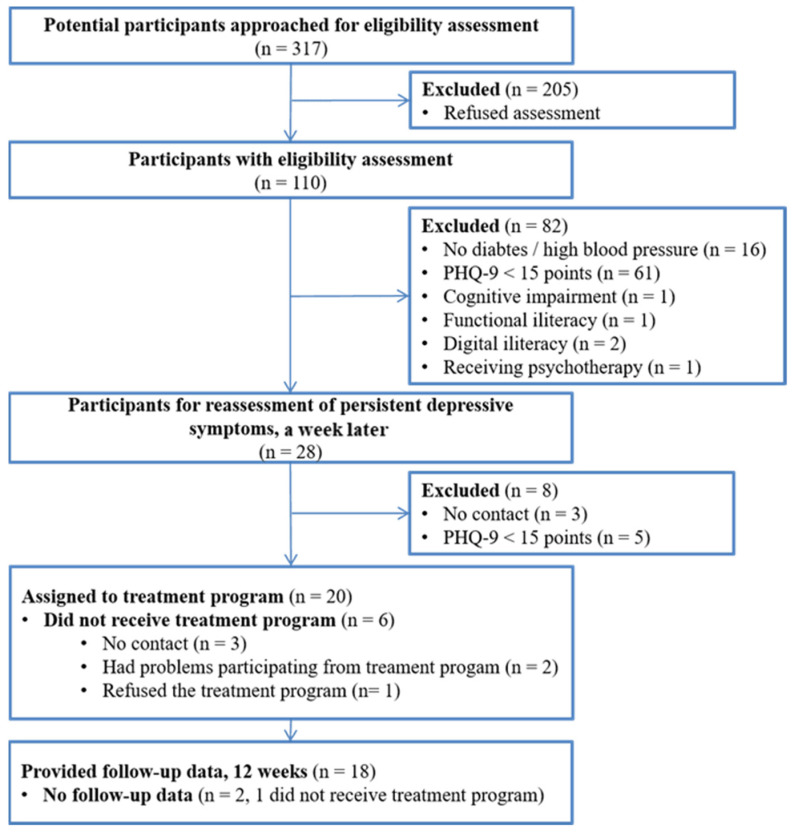
Study flow diagram.

**Figure 2 ijerph-18-12000-f002:**
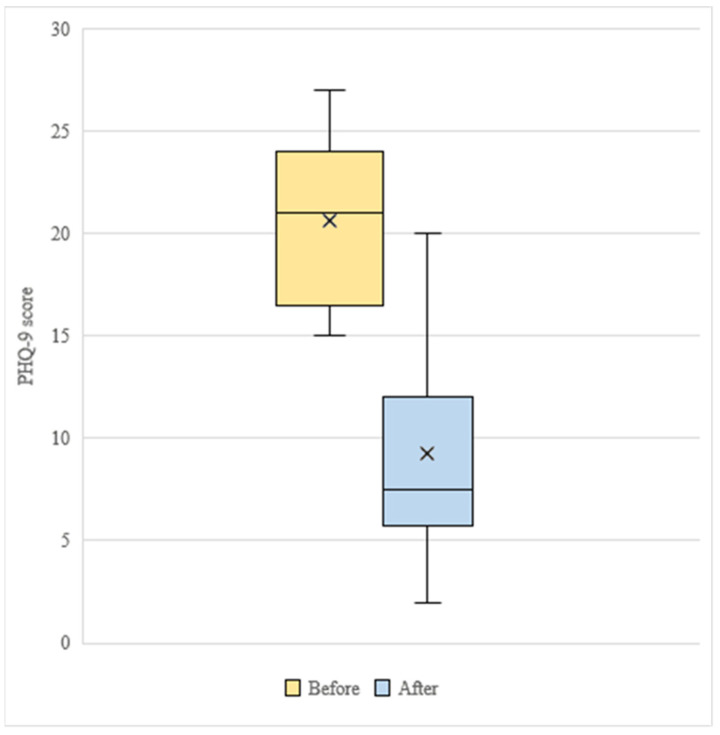
Boxplot for PHQ-9 scores before and after the intervention. Vertical lines represent medians, and crosses represent means.

**Table 1 ijerph-18-12000-t001:** Structure of the computer-assisted psychosocial intervention.

Session	Name	General Objective
1	What happens with my health?	Patients are introduced to the program and receive brief psychoeducation on depression.
2	What are diabetes and high blood pressure all about?	Patients received education on diabetes and/or high blood pressure care and their relationship with depression.
3	How to solve my problems?	Patients are introduced to problem-solving techniques.
4	Learning to solve my problems	Patients are motivated to use problem-solving techniques in their daily life.
5	What did I learn in this therapy?	Evaluate contents learned and reinforce those that are less incorporated.

**Table 2 ijerph-18-12000-t002:** Personal and family health history for the sample.

Variable	*n* (%)
**Previous diagnosis of depression**	13 (65.0)
**History of treatments for depression** ^1^	13 (100.0)
**Family history of mental health** ^2^	
Depression	12 (63.2)
Alcohol/drugs	6 (33.3)
Anxiety	5 (26.3)
Others (psychosis, bipolarity, suicide)	7 (35.0)
**Cigarrette consumption**	9 (45.0)
**Medical comorbidities** ^3^	
Arthrithis/osteoarthritis	9 (47.4)
Heart disease	3 (16.7)
Asthma/Emphysema, lung	3 (15.8)
Thyroid problems	7 (38.9)
Other medical comorbidities	11 (55.0)

^1^ The denominator corresponds to those participants who declared that they received a previous diagnosis of depression (*n* = 13); ^2^ The denominators are 20 cases for “Others”, 19 cases for “Depression” and “Anxiety”, and 18 for “Alcohol/drugs”; ^3^ The denominators are 20 cases for “Other medical comorbidities”, 19 cases for “Asthma/Emphysema of the lung” and “Arthritis/Osteoarthritis”, and 18 cases for “Heart disease” and “Thyroid problems”.

## Data Availability

The data presented in this study are available on request from the corresponding author. The data are not publicly available due to ethical reasons.

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
