# Peer review of "Technology-Assisted Collaborative Care Program for People with Diabetes and/or High Blood Pressure Attending Primary Health Care: A Feasibility Study"

_ijerph, 2021, doi:10.3390/ijerph182212000_

Round 1
Reviewer 1 Report
Thank you for a well written paper.
The study is has opportunity to inform future technology assisted clinical trials.
A small typo:
Typo in the flow chart: it should be 'a week later'?
Author Response
Dear Reviewer,
Thank you so much for your observation.
We have replaced Figure 1 with a version without the typo.
Please see the attachment.
Have a nice week.
Pablo.
Reviewer 2 Report
This is an interesting pilot study in a primary care setting that investigates the use of a collaborative care model to improve depressive symptoms in patients with diabetes and/or high blood pressure. in Chili.
The introduction is csufficiently clear, although the objective of the study could be explained in more detail. It currently states that the objective is to investigate feasibility of the CC model; however, what aspects of feasibility do the authors refer to? EWith regards to effectiveness, organisation of care, patient satisfaction, technological feasibility?
The methods are described in much detail, accept for the qualitative data analysis, which is only marginally described. However, since the ' thematic' results follow the topics discussed one on one, not much detail on the analysis approach is required. In fact, the results from the qualitative work could be reported more briefly if needed.
Figure 1 of the results section provides a lot of insight into reasons for exlclusion of patients into the study. Were the 317 initially approached subjects all potentially eligible, namely having depressive symptoms as well as depression and/or diabetes? The remaining number of patients that finally entered the study was remarcably lower. What did the authors learn from this difficult recruitment and would it be feasible to test this at a larger scale?
The final step is not fully clear to me: 20 subject were assigned to the treatment program; of which 6 were not assigned to a treatment program. However, after 12 weeks, 18 of the 14 that were left provided follow up data. How is this possible?
3.4: 'The computer-assisted psychosocial intervention achieved a global acceptability of 92.3%'. What does this ' global' acceptability mean?
The authors conclude that the program was feasible to implement. However, part of it (CONEMO) was not implemented at all, since it was not acceptable to the patient population. How does this result align with this conclusion? Please consider to re-phrase.
Author Response
Dear Reviewer,
We appreciate your insightful comments and observations.
We agree with the comment regarding the breadth of the concept of “feasibility”. We have worked under the scope proposed by Bowen et al. (“How we design feasibility studies,” ref. 48 in the manuscript). According to this framework, we assessed “acceptability” (e.g., satisfaction, intention to continue use, and perceived appropriateness), “demand” (actual use), and “potential efficacy”. We have added a brief comment on this in the “study design” subsection: “This is a feasibility study to inform the conduct of a future clinical trial, which focused on the following key areas: acceptability, demand, and potential efficacy.
As indicated in the procedures section, Figure 1 represents 317 participants who were approached in the waiting room to assess eligibility criteria, of whom only 110 agreed to answer the initial survey-which assessed eligibility criteria. Being a primary care center attended by the general population, we expected that a significant number of those willing to answer would NOT meet the eligibility criteria (see “excluded” in Figure 1). This “recruitment cascade” is typical and can be seen in the recruitment flow of previous clinical studies carried out by research team members in primary care centers in Chile. For example, in Araya et al. “Treating depression in primary care in low-income women in Santiago, Chile: a randomised controlled trial” (ref. 25 in the manuscript), where 7% of those initially evaluated are finally assigned to the intervention.
The discrepancy between subjects receiving treatment (n = 14) and subjects receiving evaluation (n = 18) is explained by applying the intention-to-treat principle. We prefer to report the outcomes of subjects independent of whether or not they received the program according to the intervention protocol. As is done in clinical trials, this allows us to obtain a less biased approximation of the potential efficacy of the intervention.
By “global acceptability” we refer to the overall acceptability perceived by the subjects, which is the compounded measure of the different dimensions included in the acceptability instrument.
We rephrased all of the statements referring to the feasibility of the intervention program, specifying that it was “partially feasible” to implement.
Please see the attachment.
Many thanks agains,
Have a nice week.
Pablo
Reviewer 3 Report
Describe why you conducted the study with the sample described in it. Are there similar studies that use samples of the same size?. Justify how the necessary sample size was determined to have sufficient statistical power and explain it with a couple of lines in the methodology section.
The format of the references must conform to the standards of the journal.
Author Response
Dear Reviewer,
Thank you so much for your comments and observations.
We agree that concerns on sample size and statistical power should be part of any study aimed at generalizing its results. However, we should mention that this is an early-phase feasibility study. Bowen et al., in “How we design feasibility studies” (ref. 48 in the manuscript), states that feasibility studies often are conducted in convenience samples with limited statistical power. Moreover, the CONSORT 2010 statement (https://www.bmj.com/content/bmj/355/bmj.i5239.full.pdf), in its extension to randomised pilot and feasibility trials indicates the following, when referring to the methodological considerations guiding the development of this statement: “formal hypothesis testing for effectiveness is not recommended. The aim of a pilot trial is not to assess effectiveness and it will usually be underpowered to do this” (p.4). As such, and understanding the scope of a feasibility study, we use the term “potential efficacy” throughout the manuscript.
For a reviewed version of the manuscript, please see the attachment.
Many thanks again.
Have a nice week!
Pablo